# The Effects of Locus of Control, Agents of Socialization and Sport Socialization Situations on the Sports Participation of Women in Taiwan

**DOI:** 10.3390/ijerph16101841

**Published:** 2019-05-23

**Authors:** Hsiu-Chin Huang, Li-Wei Liu, Chia-Ming Chang, Huey-Hong Hsieh, Hsin-Chi Lu

**Affiliations:** 1Physical Education and Arts School, Chengyi University College, Jimei University, Xiamen 361021, China; op5166@yahoo.com.tw; 2Department of Leisure Service Management, Chaoyang University of Technology, Taichung 41349, Taiwan; ijofsrm@gmail.com; 3Department of Physical Education, Health & Recreation, National Chiayi University, Chiayi 62103, Taiwan; gr5166@yahoo.com.tw; 4Department of Leisure Management, Taiwan Shoufu University, Tainan 72153, Taiwan; 5Department of Physical Education, Health & Recreation, National Chiayi University, Chiayi 62103, Taiwan; s1031021@mail.ncyu.edu.tw

**Keywords:** personal character, agents of socialization, sports socialization situations, sports participation, women

## Abstract

Compared to men, the sports participation of women is lower, especially in the East. Not many studies have compared the impacts of locus of control, agents of socialization, and sport socialization situations on the sports participation of women. Hence, the purpose of this study is to explore the contributing factors which may promote the sports participation of women in Taiwan. To do this, 450 structured questionnaires were distributed to women in Chiayi, Taiwan, with an 89.3% return rate. The study found that internal locus of control, agents of socialization, and sport socialization situation had positive impacts on the sports participation of women. In line with these results, the study suggests the strengthening of the internal locus of control of women, making the best use of socialization agents, and improvement of sport socialization situations, in order to promote sports participation in women.

## 1. Introduction

With a prevalence of sedentary lifestyles and poor dietary habits, many people suffer from obesity-induced diseases [1]. Participation in sports can promote health and help to avoid these diseases. For instance, Humphreys et al. [2] argued that regular exercise brings about the following advantages for human physical and mental health: improvement of physical structure composition, enhancement of physical function and activity, promotion of life tranquility and comfort, and increase of spiritual and intellectual activities. In addition, some studies [3,4] stated that sports participation can reduce depression and anxiety, avoid trait anxiety, and decrease mild and medium frustration. Therefore, regular engagement in sport activities can improve the physical state and promote mental health.

Woelfel and Haller [5] indicated that, in the process of sports socialization, there are three principal factors, including significant others (agents of socialization who can be imitated), sport socialization situations and opportunities (such as family, school, or community projects and facilities), and role-learning of general and personal attributes (such as personal character). Many related studies [6,7] have mentioned the effects of sports socialization and sports participation. Therefore, these agents of socialization gradually become the objects of study in research associated with the sociology of sport.

With a change in social structures and the rise of self-awareness in women, the participation of women in sport-related activities is more common than before, with some also participating in sports. For instance, in Taiwan, there are more and more female participants in running, cycling, or triathlon competitions [8]. In the past, the participation of women in sports has been rare. In the sports field, there have existed inequalities caused by traditional gender stereotypes. For instance, due to the gender stereotype of sports competence, opportunities for women to join in on sports activities and the capacity demonstrated have been limited [9]. Patriarchal society tends to define males as strong, active, and powerful beings and females as tender, passive, and obedient ones. This traditional norm is an invisible constraint for women and hinders their participation in sports [10,11].

However, practicing sports is an important tool for socialization. According to process of socialization, with the gender norm, the public has different views and standards regarding male and female sports participation, which results in the female gender role and the consequent obstacles. The process of sports socialization and the sources of its effect on socialization are extremely diverse. It influences the process for women to approach sports and leads to different personalities and behavior [12,13]. Some previous studies have revealed that the percentage of female participation in sports was low, less than males [14,15]. Hence, knowing the driving and hindering factors influencing the participation of women in sports may help to promote the engagement of women in sport. In this study, we investigated 450 female subjects in Taiwan using a structured questionnaire to explore the effects of personal character, agents of socialization, and sport socialization situations on the sports participation of women. We hope these findings can contribute to the promotion of the participation of women in sports and to further understand the inner and outer drivers of and barriers to female engagement in sports.

## 2. Research Hypotheses

### 2.1. The Relationship between Locus of Control and Sports Participation

According to Rotter’s social learning theory [16], locus of control is the belief to which people determine whether or not they get reinforced in life. People can be classified along a continuum, from very internal to very external. A person with a very internal locus of control believes the responsibility of whether he or she gets reinforcement lies with himself/herself. On the other hand, people with an external locus believe the reinforcements in life are controlled by chance, opportunity, destiny, or by powerful others. As Rotter pointed out, the locus of control is a generalized expectancy; it can predict the behavior of people across different situations. Namely, their behavior can be affected by the environment and situations they encounter. For example, Arslan et al. [17] investigated 514 college students aged 18–27 and found that students with a higher internal locus had significantly lower trait anxiety scores than those with a higher external locus of control. In relation to body image, Chang and Tsai [18] argued that the locus of control is the key factor determining the body image of an individual. Furnham and Grecves [19] also argued that people with a higher internal locus of control have higher intention to improve their body image. In comparison to those with externals, internals are happier and more satisfied with their body image [20]. According to Leonard [21] and Snyder [22], the unique social, individual, and environmental learning and experiences of an athlete results in their progress in the sport field. Therefore, we propose the following:

**Hypothesis** **1** **(H1):**
*The participation of women in sports is influenced by their locus of control.*


### 2.2. The Relationship between Agents of Socialization and Sports Participation

Socialization is a process in a group which teaches each member the rules (social norms) and skills to participate in that society. Agents who are key figures in enforcing social rules are called socializing agents [23]. Agents of socialization include family, institutional agents (school, workplace, and government), friends, the community, and mass media. Specifically, after adolescence, an institutional agent has much less influence on the sports participation of individuals and, as the interest of our study is related to the sports participation of female adults, we have excluded the effect of institutional agents. Therefore, the agents of socialization considered in the study include family, friends, mass media, and the community. Some studies have explored the effect of agents of socialization on sports participation. As for family, family encouragement and support for participation in sports would reinforce the attitude towards sports of an individual, along with participation motivation and behavior [24,25]. There is significant correlation between the participation motivation and the significant others of athletes [26]. For example, the intention of an individual to participate in sports is influenced by the support and interaction with school administrators, physical education teachers, coaches, classmates, or sports teams [27]. As for friends, Lin et al. [28] studied female teachers who joined in table tennis and found that the affective support and belonging of their friends could satisfy their sense of achievement and self-realization, which was the drive for their continuity in table tennis. As for community, in Taiwan, people can access public school gymnasiums (such as swimming pools, badminton courts, basketball courts, and so on) after-school hours and some schools even rent them out to residents, as long-term sports places, by contract in off-peak hours (after-school hours) [29]. The accessibility and convenience of sports fields in the community may also increase sports participation intention in the community. As for mass media, Chang [30] stated that, by transmission of media, many people were profoundly affected. For instance, TV broadcasts, newspaper reports, and related sports magazines or websites provide knowledge associated with sports or information related to sports games, as well as the creation of sports stars and the introduction of sports appliances. These all influence individual sports participation behaviors. Based on a previous literature review, agents of socialization are key factors in individual sports participation. Thus, this study proposes the following:

**Hypothesis** **2** **(H2):**
*Family has a significant impact on the sports participation of women.*


**Hypothesis** **3** **(H3):**
*Friends have a significant impact on the sports participation of women.*


**Hypothesis** **4** **(H4):**
*The community has a significant impact on the sports participation of women.*


**Hypothesis** **5** **(H5):**
*Mass media has a significant impact on the sports participation of women.*


### 2.3. The Relationship between Sport Socialization Situations and Sports Participation

Common social organizations or social factors of socialization learning refer to socio-economic status, background, location, and facilities [31]. Sport socialization situations refer to the convenience of sport facilities for a person to participate in sports. Chien [32] argued that the sports environment can enhance sports behavior and environmental demands, including accessibility, extensity, safety, support, and aesthetics. Yang et al. [33] found that community sports facilities are important factors for public participation in sports. For instance, Sallis and Kerr [34] proposed six items of community design which influence physical activity level, including community connectivity, density, land-use mix, aesthetics, safety (or perceived safety), and public transit. Reduction of distance between the location of an individual and the destination, to allow them to walk or ride a bike instead of driving a car and increasing their visit intention by having a safe and aesthetic environment reinforces their physical activity in daily lives. Therefore, this study proposes the following:

**Hypothesis** **6** **(H6):**
*Sport socialization situations have a significant impact on the sports participation of women.*


## 3. Methods

### Participants

Female adults, over 20 years old, were invited to participate in our study in Chiayi City and Chiayi County in Taiwan. A structured questionnaire was distributed in markets, shopping malls, department stores, post offices, and banks in Chiayi County and City where women frequently appeared. A total of 450 questionnaires were distributed. Before questionnaire distribution, subject consents were acquired. A total of 402 valid questionnaires were retrieved, with a valid return rate of 89.3%. Among all participants, 71.9% were married. The descriptive statistics also show that the most prominent educational level of respondents (46%) was a Bachelor degree. In 97% of respondents, the monthly income was under US$2600; most were between 31–40 (34.6%); most had two children (35.1%); 374 subjects (79.0%) had full-time job; 84 subjects (20.9%) participated in sports clubs; and 90 subjects (22.4%) had participated in school sports teams in recent years (see Table 1 for details).

## 4. Measurements

### 4.1. Locus of Control Scale

The extent of the locus of control of an individual can range from an extremely internal to extremely external. To measure the locus of control of respondents, we used a four-item ‘internal locus of control scale’, designed by Rotter [16], using a Likert’s five-point scale, ranging from “1 = strongly disagree” to “5 = strongly agree” (see Appendix A for details). If an internal locus of control was manifest, the total score is above 12. On the contrary, people with a higher external locus of control will obtain a lower score. Cronbach’s α of the total scale was 0.724.

### 4.2. Agents of Socialization Scale

The scale design for agents of socialization was designed upon the related literature and the “sports socialization scale”, designed by Chen et al. [27], which was revised, according to the research subjects. There were four dimensions: “family”, “friends”, “community”, and “mass media”, including 14 items. Likert’s five-point scale was used, ranging from “1 = strongly disagree” to “5 = strongly agree” (see Appendix B for details). Cronbach’s α for family was 0.821; for friends was 0.746; for community was 0.704; for mass media was 0.794; and for the total scale was 0.849.

### 4.3. Sport Socialization Situations Scale

The scale for sport socialization situations was designed using a revision based on the items of the environmental facility scale of Chien [32]; including “safety”, “extensity”, “convenience”, “atmosphere”, and four items, such as “locations of sports places I frequently visit are safe” and “locations of sports places I frequently visit are easy to reach” (See Appendix C for details). Likert’s five-point scale was used, ranging from “1 = strongly disagree” to “5 = strongly agree”. Cronbach’s α for the total scale was 0.858.

### 4.4. Level of Exercise Participation

Level of exercise participation was estimated using Fox’s Exercise Engagement Scale [35]. The scale was comprised of three items, in terms of frequency, intensity, and duration. As for frequency, respondents were asked “How often do you do exercise in a week?”, ranging from “1 = less or equal to once” to “5 = equal to or more than five times”. As for intensity, respondents were asked “On average, how did you feel after working out?”, ranging from “1 = not tired at all” to “5 = exhausted”. As for duration, respondents were asked “On average, how many minutes did you exercise each time?”, ranging from “1 = less than 30 min” to “5 = over 120 min” (See Appendix D for detail). To estimate the level of exercise participation, we used Fox’s formula as follows:Level of exercise participation = frequency ∗ (duration + intensity).

The higher the figure was, the higher the level of exercise participation. Cronbach’s α of the total scale was 0.858.

### 4.5. Data Analysis

Six hypotheses were developed in this study and data analysis was carried out using partial least squares (PLS). PLS is similar to structural equation modeling (SEM) and measures the correlation of constructs [35,36]. There are many statistical software packages that enable users to use PLS. Warp PLS 5.0, developed by Kock [37], was used in the present study.

## 5. Results

### 5.1. Descriptive Statistics

Table 2 presents the summary statistics of all construct variables. Among all prediction variables, locus of control was the highest, indicating most subjects had a high internal locus of control. The second highest was family, indicating that the family support of participants was high (M = 3.80, SD = 0.65). Sport socialization situations was the third highest, indicating that the perception of sport socialization situations was high (M = 3.77, SD = 0.65). As for sports participation, compared to the scale score range (2–50), the statistics showed that the level of sports participation was very low (M = 9.63, SD = 4.76).

Among the background variables for women about sports participation, no significant differences were found between marriage, level of education, age, the number of children, or whether they had a full-time job. There were significant differences in monthly income, whether they were members of sport clubs at the time of the study, and previous experience in participating in school sport teams during schooling.

### 5.2. The Structural Model and Hypothesis Testing

An evaluation of the structural model is used to examine the six hypothesized relationships. The test results are shown in Table 3 and Figure 1. In line with the value and significance of the path coefficients, locus of control (β_1_ = 0.19, *p* < 0.05), family (β_2_ = 0.10, *p* < 0.05), friends (β_3_ = 0.29, *p* < 0.05), and sport socialization situations (β_6_ = 0.34, *p* < 0.05) appear to have positive impacts on sport participation, while community (β_4_ = 0.04, *p* > 0.05) and mass media (β_5_ = 0.05, *p* > 0.05) do not.

Based on the path estimates, it was found that sport socialization situations was the most important predictor of sport participation (standardized path coefficient = 0.34), the second most important predictor was friends (standardized path coefficient = 0.29), and the third most important was locus of control (standardized path coefficient = 0.19).

In addition, the six variables, all together, explained a 38% variance in the sports participation of women (R^2^ = 0.38).

## 6. Discussion

According to the statistical results presented above, it seemed that the model examined in the current study was able to reach an acceptable level, in the terms of predictive power (R^2^ = 0.38). In relation to the path coefficient analyses, locus of control was found to have predicting power with a coefficient of 0.19, which indicated that women with a higher internal locus of control had a higher sport participation level. This result provides support for the significant effect of personal character on sports behavior. Crothers et al. [38] argued that people who lack internal control tend to encounter emotional exhaustion and discomfort. On the other hand, people with internal control are less stressful. In relation to the sports field, research found team players had higher positive perception of the leadership of a coach when the coach had internal control [39]. In a study of university students in table-tennis courses related to locus control, the study found that the internal control-oriented subjects who learned difficult work first had more accurate performance than the external control-oriented ones [40,41,42,43,44,45,46,47,48]. In summary, people with a higher internal locus of control had better performance in the sports-learning process and were less frustrated. They believed that they could make progress and perform well with effort. Hence, reinforcement of the internal locus of control for women is one of future directions for promoting the sports participation of women.

The effects of friends and family on sports participation of women are significant and positive. This suggests that sports participation levels can be enhanced by friends and family. These results were supported by previous findings. For example, Coleman et al. [41], who interviewed seventy-five 15–19 year-old young British women, realized that the friendship group was the most critical factor in sports participation. Young et al. [42] argued that, when sports are supported by friends, the interactive quality and pleasure are enhanced. Sharma et al. [43] stated that family member support reinforced the sports participation of African American women in their leisure time. Therefore, the company of friends and the encouragement of family member increase the sports participation intention of women. On the other hand, the effects of the community and mass media on sports participation of women were not significant. It seemed that support from the community (nearby sport field, sport-related events, and the company of neighbors) for sports participation was not strong enough. Additionally, support from the mass media was not significant, indicating the necessity of posting more female-oriented sports-related agenda, events, and advertisements.

Female sport socialization situations significantly and positively influence sports participation. This is consistent with previous research findings, which argued that convenience of environment for sports socialization significantly influenced sports participation. For instance, people tend to practice running and walking when there is park in the neighborhood. Dirtiness in sports places, such as too much trash and the excrement of animals, lowers their sports intention [49]. Vojnovic [45] stated that, when the distance to sports place was less than 0.8 km, it encouraged those who could walk to the place to practice exercises; and sports places with distances less than 5 km attracted bike riders. Liu et al. [46] stated that the spatial design of sports places and sports facilities influenced intention for sports participation. For instance, when the lighting of sports places is sufficient and the width of running tracks is appropriate, participants perceive comfort and safety, which, in turn, can attract people to go there for exercise. Duncan [47] suggested that the crime rate and the problem of stray dogs in sports places negatively affected intention of sports participation. Therefore, we suggest the enhancement of environmental atmosphere, convenience, and safety to promote the sports participation of women.

## 7. Recommendations

Based on the findings of this study, for women in Chiayi County and City, sport socialization situations are the most important key factor for the sports participation of women; the second most important factor is friends (as agents of socialization); the third most important is locus of control; and the fourth is family (as agents of socialization).

As for sport socialization situations, we suggest that related governmental or administrative sectors can plan sports places for women, to allow them to have comfortable and safe sports environments. In addition, convenience is one of the key factors. If sports places are too distant, the women will not be motivated. Therefore, sports places for women can be installed in the downtown or surrounding areas, to enhance convenience of location and create a positive and comfortable sports environment and atmosphere. In addition, they can plan sports programs exclusively for women, such as running and sports meet competitions, to reinforce their sports participation.

As for agents of socialization, we suggest women to practice sports with family members or friends, and change shopping or afternoon-tea time into mountain climbing or hiking time; not only to enhance their physical strength, but also to increase their pleasure in sports and techniques.

Locus of control is another factor affecting the sports participation of women. Therefore, we recommend the government to promote equal education opportunities and encourage women to engage in sports, to increase the internal locus of control of women toward sports participation.

The results of this study revealed that “community” and “mass media”, as agents of socialization, are not associated with the sports participation of women. This result differs from some related research [48,49]. It is suggested that, in future research, the reason for this should be explored, in order to further explore the effects of personal character and sports agents of socialization on the sports participation of women and to compare the cultural differences among nations.

## Figures and Tables

**Figure 1 ijerph-16-01841-f001:**
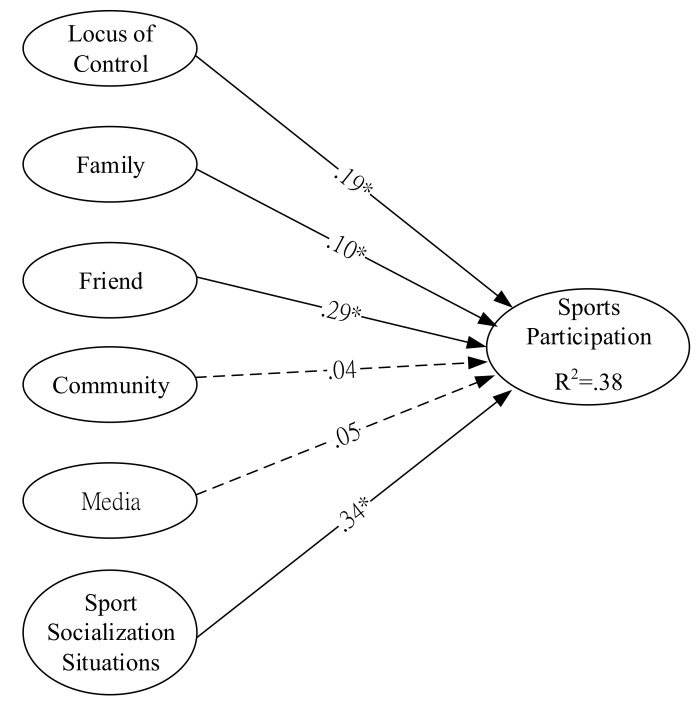
Structural equation modeling (SEM) results of the standardized model parameter estimation. * *p* < 0.05; solid line: significant; dashed line: non-significant.

**Table 1 ijerph-16-01841-t001:** Description of the demographic variables of the participants.

Variables	Groups	N	%	Variables	Groups	N	%
Education Level	Elementary school	5	1.2	Marital status	Married	289	71.9
	Junior high school	9	2.2		Unmarried	113	28.1
	Senior high school	57	14.2	Monthly income (USD)	No income	15	3.7
	Junior college	75	18.7		Between $1 and $650	25	6.2
	College	185	46.0		Between $650 and $1300	184	45.8
	Graduate school	71	17.7		Between $1300 and $2600	166	41.3
Age	Between 20 and 30 years old	77	19.2		Exceeded $2600	12	3.0
	Between 31 and 40 years old	139	34.6	Number of children	No children	131	32.6
	Between 41 and 50 years old	130	32.3		One	62	15.4
	Between 51 and 60 years old	44	10.9		Two	141	35.1
	Exceeded 61 years old	12	3.0		Three or more than three	68	16.9
Full time job	Yes	374	93.0	Participate in sport clubs	Yes	84	20.9
	No	28	7.0		No	318	79.1
Participation in school sports teams in past years of study	Yes	90	22.4				
	No	312	77.6				

**Table 2 ijerph-16-01841-t002:** Descriptive statistics of all construct variables.

Variables	Mean	Standard Deviation	Score Range
LC	4.11	0.57	1–5
FA	3.8	0.65	1–5
FR	3.53	0.66	1–5
COM	3.04	0.78	1–5
MM	3.07	0.69	1–5
SSS	3.79	0.65	1–5
SP	9.63	4.76	2–50

Note: LC: locus of control; FA: family; FR: friend; COM: community; MM: mass media; SSS: sport socialization situations; SP: sport participation.

**Table 3 ijerph-16-01841-t003:** Path results for the structural model.

Hypothesis	Path	Path Coefficient	*p*-Value
H1	LC → SP	0.19 (β_1_)	<0.05 *
H2	FA → SP	0.10 (β_2_)	<0.05 *
H3	FR → SP	0.29 (β_3_)	<0.05 *
H4	COM → SP	0.04 (β_4_)	>0.05
H5	MM → SP	0.05 (β_5_)	>0.05
H6	SSS → SP	0.34 (β_6_)	<0.05 *

Note: * significant; LC: locus of control; FA: family; FR: friends; COM: community; MM: mass media; SSS: sport socialization situations; SP: sport participation.

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
