# Peer review of "The Effects of Locus of Control, Agents of Socialization and Sport Socialization Situations on the Sports Participation of Women in Taiwan"

_ijerph, 2019, doi:10.3390/ijerph16101841_

Round 1

Reviewer 1 Report

Greetings!

 The topic is a worthwhile endeavor and it is exciting to see researchers taking a more in depth look at female sport participation. The paper addresses many issues which are important to understanding this issue yet more depth is needed tie the roles these issues play in the current state of female sport participation. 

I believe this work needs to be published with more attention to addressing the previous barriers in a clearer way. My suggestions:

* Review for grammar (subject/verb agreement, verb tenses). Example: In the abstract it says the "purposes" but only one is stated.

Introduction

*  The first line speaks to technology and economy as factors of health. How does sport address these factors to promote health? I suggest taking this line out as it does not get at the crux of your argument in the present manuscript or providing more context to why these factors shape sport participation. 

* Be clear about your terminology. In the paper sport, exercise, and leisure activity are used but these can be considered different things.   What are the elements of sport that bring about the health benefits described.

* The sentence stating "Many related studies." needs citations. 

* How does gender affect sport participation? This where you can link back to the technology and economic factors still falling into social patterns of inequality due to current social structures and misogynistic ideas about women.

Hypothesis

* Family, Media, Friends, and social structures are given a light treatment. I suggest having a clear section describing each factor then stating the related hypothesis. This will strengthen your results and your model as it will justify each factor as being worthy of individual investigation (even when two were proven statistically insignificant, you can still show the practical importance and need to investigate those factors in a different way)

* For sport situations, many aspects of the sport situation are stated without further explanation. I suggest focusing on how the sport situation is conceptualized in this study and why this definition was chosen. (i.e. described what is safety, extensity, convenience, etc. and how it affects sport participation)

* For the hypotheses, I suggest rephrasing by removing the "Agents of socialization' as this should be understood from the section/ justification proceeding it. 

Method

* The word "treated" suggest this is an experiment. I suggest rephrasing. 

* Descriptive statistics were conducted? Your results could also speak to the intersection of gender and other social factors (economic status, education, family status) if this analysis is provided.

Discussion/ Recommendations

* I would speak to the areas that can be controlled/ changed as that would provide new insight to this topic. I believe this will also lead to more structured (implementable) recommendations. 

* Recommendations should consider the barriers that kept these advancements from being implemented for this population (particularly the gender ideologies shaping women's sport participation).

I believe this study can have a positive contribution for enhancing women's sport particiaption but we must be careful to lay out how so that  the recommendations do not fall in the same trap of being ineffective by not being comprehensive (see Shaw and Frisby (2006): 

Can Gender Equity Be More Equitable?: Promoting an Alternative Frame for Sport Management Research, Education, and Practice

Author Response

The topic is a worthwhile endeavor and it is exciting to see researchers taking a more in depth look at female sport participation. The paper addresses many issues which are important to understanding this issue yet more depth is needed tie the roles these issues play in the current state of female sport participation. 

I believe this work needs to be published with more attention to addressing the previous barriers in a clearer way. My suggestions:

* Review for grammar (subject/verb agreement, verb tenses). Example: In the abstract it says the "purposes" but only one is stated.

 Ans: Thank you for the comment. We had carefully checked the grammars and did spelling checks as well.

Introduction

*  The first line speaks to technology and economy as factors of health. How does sport address these factors to promote health? I suggest taking this line out as it does not get at the crux of your argument in the present manuscript or providing more context to why these factors shape sport participation. 

  Ans: Thank you for the comment. We deleted the phrase and added one reference and the first two sentences had changed to “With prevalence of sedentary life styles and poor dietary habits, many people suffer from obesity-induced diseases [1]. Sports can promote health and avoid the diseases.”

* Be clear about your terminology. In the paper sport, exercise, and leisure activity are used but these can be considered different things.   What are the elements of sport that bring about the health benefits described.

  Ans: Thank you for the comment. We had re-write this section and focusing on the benefits from sports participations.

* The sentence stating "Many related studies." needs citations. 

   Ans: Thank you for the comment. We added two references to support this argument.

* How does gender affect sport participation? This where you can link back to the technology and economic factors still falling into social patterns of inequality due to current social structures and misogynistic ideas about women.

 Ans: Thank you for the comment.

Hypothesis

* Family, Media, Friends, and social structures are given a light treatment. I suggest having a clear section describing each factor then stating the related hypothesis. This will strengthen your results and your model as it will justify each factor as being worthy of individual investigation (even when two were proven statistically insignificant, you can still show the practical importance and need to investigate those factors in a different way)

 Ans: Thank you for the comment. We rewrote the section to make it clear. 

* For sport situations, many aspects of the sport situation are stated without further explanation. I suggest focusing on how the sport situation is conceptualized in this study and why this definition was chosen. (i.e. described what is safety, extensity, convenience, etc. and how it affects sport participation)

* For the hypotheses, I suggest rephrasing by removing the "Agents of socialization' as this should be understood from the section/ justification proceeding it. 

  Ans: Thank you for the comment. We rewrote the section to make it clear. 

Method

* The word "treated" suggest this is an experiment. I suggest rephrasing. 

  Ans: Thank you for the comment. We rewrote the sentences as “Female adults over 20 years old were invited to participate in our study in Chiayi City and Chiayi County of Taiwan.”

* Descriptive statistics were conducted? Your results could also speak to the intersection of gender and other social factors (economic status, education, family status) if this analysis is provided.

  Ans: Thank you for the comment. We conducted correlation analyses.

Discussion/ Recommendations

* I would speak to the areas that can be controlled/ changed as that would provide new insight to this topic. I believe this will also lead to more structured (implementable) recommendations. 

* Recommendations should consider the barriers that kept these advancements from being implemented for this population (particularly the gender ideologies shaping women's sport participation).

 Ans. Thank you for the suggestion. We had rearranged the presentation so to include the effect of locus of control.

I believe this study can have a positive contribution for enhancing women's sport particiaption but we must be careful to lay out how so that  the recommendations do not fall in the same trap of being ineffective by not being comprehensive (see Shaw and Frisby (2006): 

Can Gender Equity Be More Equitable?: Promoting an Alternative Frame for Sport Management Research, Education, and Practice

Reviewer 2 Report

The manuscript provided by the authors explores the factors that influence the sports participation of women in Taiwan. The study builds upon the data collected via structured questionnaires. Data analysis reveals as the factors most prominently promoting women’s sports participation in Taiwan are sport socialization situations (i.e., facilitating environments and locales for sports), socialization agents (friends and family) and internal locus of control.

Broad comments

Headline: The name of the manuscript is informative and concise. The authors should still consider adding some specificity to the headline, i.e., … Sports Participation of Women in Taiwan.

Abstract: The abstract provides information about the purpose, the method of gathering data, the results, and the recommendations drawn from the results of the study. The abstract should also have, at least, the information about the analysis method used. Furthermore, the originality of this specific study should be highlighted (i.e., what new information this study provides when compared to earlier studies). As a suggestion, the results provided in the abstract could be somewhat more specifically stated by the authors.  There are also some minor grammar issues (i.e., ”In line with the results, the study suggest to strength women’s….” to “…the study suggests to strengthen women’s…”)

The originality of the study: This might be one of the biggest omissions in this manuscript. The authors must present and highlight the novelty and originality of their study. What new information does the study offer (that we don’t already know)? Maybe the authors could use these following questions when considering the novelty of their study: Why this research setting? Does the authors’ study provide some knowledge that fills some current research/knowledge gap? Why is this study important or timely academically/societally? (i.e., the perspective of public health promotion)

English language: For the most part, the English language used in the manuscript is understandable. There are, however, grammar issues, some smaller some greater throughout the manuscript that should be addressed. (I have highlighted some of these in the specific comments section below). I would encourage the authors to have their manuscript proofread by some native English speaker.

Questionnaire: It would be strongly advisable for the authors to include their questionnaire along with the manuscript (as an Appendix or in the form of some Supplement material). This is, in my opinion, in accordance with the universal principles of scientific principles: repeatability and transparency, etc.

Measurement: It would be advisable for the authors to include summarizing figures for every scale used in this study (Locus of Control Scale; Agents of Socialization Scale; Sport Socialization Scale; Sports Participation). Authors, please provide these or give a rebuttal to this request.

References: There are some references and use of those references by the authors that should be thoroughly revised (i.e., reference [32] on page 6).

More specific comments (section by section comments)

Unfortunately, the pdf-file didn’t have line numbering so I have to refer mostly to page numbers and paragraphs.

(I)                  Introduction:

(1)    The very first sentence in the introduction states that “people suffer from civil illness”. What is meant by “civil illness”? Please specify and explain how does it relate to the health benefits of sports mentioned in the paragraph.

(2)    The second paragraph of the introduction (page 1): Are the “agents of socialization” mentioned after the “Therefore” to be identified as the same as the “three principal factors” according to Woelfel and Haller [4] “important others” ; “sport socialization situations and opportunities” ; “role learning of general and personal attributes”?

(3)    The second paragraph of the introduction (page 1): “Many related studies have mentioned the effects of sports socialization and sports participation.” The authors should provide some relevant scientific references here!

I would urge the authors to check, for instance, the following references: (1) Chen et al. (2017) Factors influencing interest in recreational sports participation and its rural-urban disparity. PLoS ONE 12(5): e0178052. https://doi.org/10.1371/ journal.pone.0178052 ; (2) Chen & Huang A Study of Sports Consciousness of Residents in Taiwan (http://thesportjournal.org/article/a-study-of-sports-consciousness-of-residents-in-taiwan/) ; (3) Tsai et al. (2015) Gender Differences in Recreational Sports Participation among Taiwanese Adults. Int J Environ Res Public Health, 12(1), 829-840, doi: 10.3390/ijerph120100829 ; (4) Women, Sport and Exercise in the Asia-Pacific Region: Domination, Resistance, Accommodation (Molnar, Amin, & Kanemasu (eds.); Routledge 2018). These sources could be useful in other sections of this manuscript too.

(4)    I cannot seem to locate the publication or the journal behind reference [5]. Please, provide URLs or DOIs with all the References as is customary!

(5)    The third paragraph of the introduction (page 2) mentions the negative effects of gender stereotypes and patriarchal thinking on women’s sports participation. Do these reflections surface in the discussion section of the manuscript? Furthermore, are there any findings from this study that could shed some light on these issues mentioned.

(6)     The final paragraph of the introduction (page 2): The last two sentences of the paragraph: are they intended as a description of the originality of this study. If so, then the authors should highlight it more explicitly (and in the light of previous studies).

(II)                Hypothesis (plural form is hypotheses) (maybe the best header would be “Research hypotheses”)

(7)    The paragraph “The relationship between locus of control and sports participation” (page 2): Should ”personal character” within this study be defined even more explicitly in terms of locus of control? See also the sections Discussion and Recommendations

(8)    The paragraph “The relationship between locus of control and sports participation” (page 2): The authors use the following terms: “personal physical image”, “figures”, “physical image” in this paragraph. Do all these terms refer to body image; if they do it would be better to simply use body image in every instance.

(9)    The paragraph “The relationship between locus of control and sports participation” (page 2): The sentence ending in reference [18] seems to be based on a study investigating teachers’ locus of control. Should the sentence then be re-written so that this becomes clear to the reader (This same principle applies also elsewhere in the manuscript: the authors should be clearer and more transparent when necessary and when referring to a study conducted, for example, with teachers or of children of a certain age, and then generalizing the results of concerning all people.

(10) The hypothesis  H1 could be “taken out” or indented from the rest of the text as is done with other five hypotheses. There could also be a “table of hypotheses” before the method section: it might bring some clarification for the reader. I’ll leave this to the authors’ discretion.

(11)The paragraph “The relationship between agents of socialization and sports participation (page 3) second line: What is meant by the authors by “They are the third parties which influence…”? It might be a reference to the “three major socialization factors” mentioned in the previous sentence. However, this expression somewhat awkward.

(12)The paragraph “The relationship between agents of socialization and sports participation (page 3):

The authors write: “Subjects of this study are all women. Therefore, agents of socialization refer to family, friends, mass media and community.” Why and on what grounds do the authors choose or narrow as the agents of socialization in women particularly these four (family, friends, mass media and community? How is this decision justified (i.e., do men have some other agents of socialization)? At least, the “therefore”, starting the second sentence seems to indicate that the four agents of socialization are specifically and exclusively pertaining only women.

(13)The paragraph “The relationship between agents of socialization and sports participation (page 3):

The authors write: “Hung [26] argued that in Taiwan, many schools make gymnasium open to community”. Is the intention of the authors rather say that “many schools should make gymnasium open”?

(14)The paragraph “The relationship between agents of socialization and sports participation (page 3):

From the agents of socialization used in this study and defined in this paragraph, the community seems somewhat vaguely defined. The authors should clarify and sharpen the concept of the community used here.

(15)The paragraph “The relationship between agents of socialization and sports participation (page 3):

Before the listing of hypotheses H2-H5 there is the sentence: “Based on previous literature review, agents of socialization are key factors of individuals’ sports participation”. What is the specific reference here? There is no reference mentioned. If “based on previous literature review” refers distinctly to some literature review(s) not mentioned in this manuscript additional reference is needed; if it refers to studies mentioned earlier in the manuscript, it is poor English language and should be revised accordingly.

(16)The last sentence in page 3 reads: ”However, this study adopts safety, extensity, convenience and atmosphere as measurement of sport socialization situations.” Why these four are singled out as the measurement dimensions of sport socialization situations in this study? This might be a good thing to explicate by the authors. Furthermore, are all these dimensions reflected upon in the results/discussion/recommendation section of this manuscript (if necessary)?

(III)              Method

(17)The paragraph ”Participants” (page 4) lines 4-5: ”Before questionnaire distribution, subjects’ intention was inquired. What is meant by the ”subjects’ intention” by the authors here? Is this a reference to a voluntary agreement to participate in research? If so, then informed consent would be the proper term.

(18)The paragraph ”Participants” (page 4): There are some awkward expressions used in this section when reporting the ”basic information” such as: ”are the most”; ”is the most” that should be corrected.

(19)In what way (if any) are the basic information/demographic information given in this section and in Table 1 (page 5) made use of in this study. This should at least be reflected upon in the discussion section of the manuscript (for example, how this data could have been used or could be used in similar future studies or can possibly be used by the authors in the future).

(IV) Measurement

(20) Table 1 (page 5): Please correct ”One children” to ”One child”

(21)The paragraph “Sports Participation (page 6): The first sentence “Sports participation aims to test participants’ average weekly frequency to participate in sports (frequency), average participation time of each time (participation time) and sports intensity”. The phrases in parentheses (frequency) and (participation time) seem redundant in this sentence. Maybe the sentence could read: Sports participation aims to test participants’ average weekly frequency to participate in sports, average participation time of each sports exercise, and sports intensity-

(22)In this same paragraph, the authors refer to Fox’s study: “According to formula applied by Fox [32] sports participation degree = frequency X (sports intensity + participation time).” This study referenced by the authors has no explicit formula of such kind presented. Fox does speak about “different formulas of frequency, intensity and duration of exercise apply for different mechanisms and perhaps different populations.” (Fox 1999, 416), but seems not to offer any explicit formula to which the authors in this manuscript refer to. Authors, please provide an explanation/revision to this or a rebuttal.

(V) Results

(23)When reporting the results (page 6) “The structural model and hypothesis testing” there could be added – especially in the case of H4 and H5 – “statistically insignificant”. Mere categories of significant and insignificant are weaker.

(24)Paragraph “Explanatory Power” (page 7): Here the authors state that the predictive power of their research model can explain 38% of the variance of women’s sports participation. In my opinion, the authors should more clearly in this paragraph state that the agents of socialization of community and mass media are in themselves statistically insignificant and how that relates to this models’ explanatory power. Please, check also the grammar in this paragraph.

(VI) Discussion

(25)First paragraph (page 8): ”Therefore, women with higher internal locus of control are not afraid of frustration and pressure in sports learning process.” The word ”therefore” here used by the authors refer to some conclusion. It is somewhat unclear, however, where this conclusion is drawn from: from this study’s results, from other studies mentioned just before this therefore-sentence. Furthermore, and perhaps even more importantly the authors should ponder the legitimacy of their conclusion or at least state this differently.

(26)Second paragraph (page 8): Why are the agents of socialization presented in the order of ”family and friends” while the results indicate the order of friends and family?

(27)Second paragraph (page 8): ”It supports that individuals’ sports participation is accomplished by agents of socialization [39].” This sentence should be rephrased: ”it supports” should take note that these are the results of this study; the reference [39] is a study conducted with young children – this should also become clear from the text.

(28)Second paragraph (page 8): The authors state that  “the finding of this study suggests that support of family and friends positively influences women’s sports participation.” In the very next sentence, they refer to the study conducted by Coleman, Cox, and Roker [40] and state that the “friendship group is the most critical factor of sports participation”. Maybe the order of presentation in the first sentence should be reversed to “friends and family” instead of “family and friends” because that is the order of the predictive power of friends (.29) and family (.10). This would be more coherent.

(29)The last whole paragraph on page 8: There are some English editing needed here: i.e., “it encouraged those who could walk to the place to practice exercises”; “For instance, when lighting of sports places is sufficient and width of path is appropriate (please specify what is meant with the “width of path” in this particular instance) ; “Therefore, in order to strengthen women’s sports participation, it must attempt to enhance (to whom does “it” refer to).

(30)A more general comment: The whole discussion section could be re-arranged so that the structure follows according to the individual correlations and impacts on women’s sports participation (i.e., (1) Sport Socialization Situations… I leave this to the authors’ discretion.

 Anyhow, the significance and “impact factors” should be clear in the text. Furthermore, there should be some deeper contemplation upon why within this research setting and based upon this data the significances of community and mass media stayed statistically insignificant (although this topic is mentioned in Recommendations)

(31)Another more general comment: In this current manuscript there is very little discussion regarding the limitations of this study as well as the possible problems of (external) validity and generalizability of the results presented by the authors. Some notions regarding the most common possible pitfalls of questionnaire-based studies could be added.

Especially, the generalizability aspect of this study’s results should be reflected upon and explicated here and also in the recommendations section (i.e., How well are the results representative of larger population/other countries/contexts, etc.? What are the limitations of the results and recommendations of this study?).

(VII) Recommendation

(32) It would perhaps be more logical to present the recommendations in the order of relevance based on the findings of this study, at least inside the categories; currently, the authors write (page 8) “agents of socialization of family, friends and sport socialization are key factors…”.  So, sport socialization situations are the most important key factor, second most important, from the category of agents of socialization are friends and then family, and locus of control come before parents. Therefore, the authors should consider revising the order of presentation here.

(33)Furthermore, “the locus of control” aspect is not very clearly presented in the recommendations section. This should be revised because of its importance for sports participation according to the findings presented by the authors.

(34)The last paragraph (page 9): “It is different from some related research”. Here the authors speak about the role of community and mass media with respect to women’s sports participation. Some references to what these “some related research” are, is required by the authors.

Some recommended resources are: Tsai’s (2009) Media systems and their effects on women's sport participation in Taiwan. Sport, Education and Society, 14:1, 37-53, DOI: 10.1080/13573320802615023 ; Chen et al. (2004) Cultural and Social Factors Affecting Women’s Physical Activity Participation in Taiwan. Sport, Education and Society, 9(3), 379-393  

Author Response

The manuscript provided by the authors explores the factors that influence the sports participation of women in Taiwan. The study builds upon the data collected via structured questionnaires. Data analysis reveals as the factors most prominently promoting women’s sports participation in Taiwan are sport socialization situations (i.e., facilitating environments and locales for sports), socialization agents (friends and family) and internal locus of control.

Broad comments

Headline: The name of the manuscript is informative and concise. The authors should still consider adding some specificity to the headline, i.e., … Sports Participation of Women in Taiwan.

Ans: thank you for the suggestion. We added “in Taiwan” to the title.

Abstract: The abstract provides information about the purpose, the method of gathering data, the results, and the recommendations drawn from the results of the study. The abstract should also have, at least, the information about the analysis method used. Furthermore, the originality of this specific study should be highlighted (i.e., what new information this study provides when compared to earlier studies). As a suggestion, the results provided in the abstract could be somewhat more specifically stated by the authors.  There are also some minor grammar issues (i.e., ”In line with the results, the study suggest to strength women’s….” to “…the study suggests to strengthen women’s…”)

 Ans. Thank you for the comment. We corrected the grammar error.

The originality of the study: This might be one of the biggest omissions in this manuscript. The authors must present and highlight the novelty and originality of their study. What new information does the study offer (that we don’t already know)? Maybe the authors could use these following questions when considering the novelty of their study: Why this research setting? Does the authors’ study provide some knowledge that fills some current research/knowledge gap? Why is this study important or timely academically/societally? (i.e., the perspective of public health promotion)

 Ans. Thank you for the comment. We had stress the importance of our research in the abstract.

English language: For the most part, the English language used in the manuscript is understandable. There are, however, grammar issues, some smaller some greater throughout the manuscript that should be addressed. (I have highlighted some of these in the specific comments section below). I would encourage the authors to have their manuscript proofread by some native English speaker.

 Ans: Thank you for the comment. We made corrections accordingly.

Questionnaire: It would be strongly advisable for the authors to include their questionnaire along with the manuscript (as an Appendix or in the form of some Supplement material). This is, in my opinion, in accordance with the universal principles of scientific principles: repeatability and transparency, etc.

 Ans: Thank you for the comments. We rewrote those sections.

Measurement: It would be advisable for the authors to include summarizing figures for every scale used in this study (Locus of Control Scale; Agents of Socialization Scale; Sport Socialization Scale; Sports Participation). Authors, please provide these or give a rebuttal to this request.

Ans: Thank you for the comments. We had rewrote those sections.

References: There are some references and use of those references by the authors that should be thoroughly revised (i.e., reference [32] on page 6).

Ans: Thank you for the comment. We had revised those. We added and deleted some irrelevant and inappropriate.

More specific comments (section by section comments)

Unfortunately, the pdf-file didn’t have line numbering so I have to refer mostly to page numbers and paragraphs.

(I)                  Introduction:

(1)    The very first sentence in the introduction states that “people suffer from civil illness”. What is meant by “civil illness”? Please specify and explain how does it relate to the health benefits of sports mentioned in the paragraph.

 Ans: Thank you for the comment. We rewrote the sentences.

(2)    The second paragraph of the introduction (page 1): Are the “agents of socialization” mentioned after the “Therefore” to be identified as the same as the “three principal factors” according to Woelfel and Haller [4] “important others” ; “sport socialization situations and opportunities” ; “role learning of general and personal attributes”?

 Ans: Thank you for the comment. We rewrote the sentences.

(3)    The second paragraph of the introduction (page 1): “Many related studies have mentioned the effects of sports socialization and sports participation.” The authors should provide some relevant scientific references here!

 Ans: Thank you for the comment. References were added.

I would urge the authors to check, for instance, the following references: (1) Chen et al. (2017) Factors influencing interest in recreational sports participation and its rural-urban disparity. PLoS ONE 12(5): e0178052. https://doi.org/10.1371/ journal.pone.0178052 ; (2) Chen & Huang A Study of Sports Consciousness of Residents in Taiwan (http://thesportjournal.org/article/a-study-of-sports-consciousness-of-residents-in-taiwan/) ; (3) Tsai et al. (2015) Gender Differences in Recreational Sports Participation among Taiwanese Adults. Int J Environ Res Public Health, 12(1), 829-840, doi: 10.3390/ijerph120100829 ; (4) Women, Sport and Exercise in the Asia-Pacific Region: Domination, Resistance, Accommodation (Molnar, Amin, & Kanemasu (eds.); Routledge 2018). These sources could be useful in other sections of this manuscript too.

(4)    I cannot seem to locate the publication or the journal behind reference [5]. Please, provide URLs or DOIs with all the References as is customary!

 Ans: Thank you for the comment.  We deleted the sentences and the related reference.

(5)    The third paragraph of the introduction (page 2) mentions the negative effects of gender stereotypes and patriarchal thinking on women’s sports participation. Do these reflections surface in the discussion section of the manuscript? Furthermore, are there any findings from this study that could shed some light on these issues mentioned.

 Ans: Thank you for the comment. We rewrote the sentences.

(6)     The final paragraph of the introduction (page 2): The last two sentences of the paragraph: are they intended as a description of the originality of this study. If so, then the authors should highlight it more explicitly (and in the light of previous studies).

 Ans: Thank you for the comment. We rewrote the sentences.

(II)                Hypothesis (plural form is hypotheses) (maybe the best header would be “Research hypotheses”)

 Ans: Thank you. We changed the subheading into Research hypotheses

(7)    The paragraph “The relationship between locus of control and sports participation” (page 2): Should ”personal character” within this study be defined even more explicitly in terms of locus of control? See also the sections Discussion and Recommendations

 And: Thank you for comments. We rewrote the sentences and deleted “personal character”. We started to explain the origin of the theory of locus of control and explain the characteristics of internals and externals

(8)    The paragraph “The relationship between locus of control and sports participation” (page 2): The authors use the following terms: “personal physical image”, “figures”, “physical image” in this paragraph. Do all these terms refer to body image; if they do it would be better to simply use body image in every instance.

 Ans: Yes, we agreed. We use “body image” to avoid confusions.

(9)    The paragraph “The relationship between locus of control and sports participation” (page 2): The sentence ending in reference [18] seems to be based on a study investigating teachers’ locus of control. Should the sentence then be re-written so that this becomes clear to the reader (This same principle applies also elsewhere in the manuscript: the authors should be clearer and more transparent when necessary and when referring to a study conducted, for example, with teachers or of children of a certain age, and then generalizing the results of concerning all people.

 Ans; We replaced the reference with another and also described the setting of the research and pointed out the outcomes.

(10) The hypothesis  H1 could be “taken out” or indented from the rest of the text as is done with other five hypotheses. There could also be a “table of hypotheses” before the method section: it might bring some clarification for the reader. I’ll leave this to the authors’ discretion.

 Ans: Thank you for the suggestion. We took out H1 and put it on the next line to be consistent with other hypotheses.

(11)The paragraph “The relationship between agents of socialization and sports participation (page 3) second line: What is meant by the authors by “They are the third parties which influence…”? It might be a reference to the “three major socialization factors” mentioned in the previous sentence. However, this expression somewhat awkward.

 Ans; Thank you for the suggestion. We rewrote this section starting by explaining the meaning  and types of agents of socialization.

(12)The paragraph “The relationship between agents of socialization and sports participation (page 3):

The authors write: “Subjects of this study are all women. Therefore, agents of socialization refer to family, friends, mass media and community.” Why and on what grounds do the authors choose or narrow as the agents of socialization in women particularly these four (family, friends, mass media and community? How is this decision justified (i.e., do men have some other agents of socialization)? At least, the “therefore”, starting the second sentence seems to indicate that the four agents of socialization are specifically and exclusively pertaining only women.

 Ans: Thank you for the comment. Agents of socialization include family, institutional agent (school, workplace, and government), friends, mass media and community. Since the interest of our study topic is related to sports participation of female adults, we had excluded the effect of institutional agent. Therefore, agents of socialization here in the study include family, friends, mass media and community.

(13)The paragraph “The relationship between agents of socialization and sports participation (page 3):

The authors write: “Hung [26] argued that in Taiwan, many schools make gymnasium open to community”. Is the intention of the authors rather say that “many schools should make gymnasium open”?

Ans: thank you for the comment. We rephrased the sentence. The sentence actually means that people can access sports courts freely after school hours and with this bonus, some people are more willingly to participate in sports after school hours or weekend.

(14)The paragraph “The relationship between agents of socialization and sports participation (page 3):

From the agents of socialization used in this study and defined in this paragraph, the community seems somewhat vaguely defined. The authors should clarify and sharpen the concept of the community used here.

 Ans: thank you for the comment. We had explained this in previous point.

(15)The paragraph “The relationship between agents of socialization and sports participation (page 3):

Before the listing of hypotheses H2-H5 there is the sentence: “Based on previous literature review, agents of socialization are key factors of individuals’ sports participation”. What is the specific reference here? There is no reference mentioned. If “based on previous literature review” refers distinctly to some literature review(s) not mentioned in this manuscript additional reference is needed; if it refers to studies mentioned earlier in the manuscript, it is poor English language and should be revised accordingly.

 Ans: thank you for the comment. We added “Popenoe, D. Sociology. Prentice Hall, NJ, USA, 1999.” as reference.  In this book, he well explained the meaning, types of agents of socialization.

(16)The last sentence in page 3 reads: ”However, this study adopts safety, extensity, convenience and atmosphere as measurement of sport socialization situations.” Why these four are singled out as the measurement dimensions of sport socialization situations in this study? This might be a good thing to explicate by the authors. Furthermore, are all these dimensions reflected upon in the results/discussion/recommendation section of this manuscript (if necessary)?

 Ans:  thank you for the comment. We had deleted the whole sentence since it made no sense. In addition, we had explained the meaning of sport socialization situations to make it more clearly to the audience. (Sport socialization situations refer to the convenience of sport facilities for a person to participate in sports.)

(III)              Method

(17)The paragraph ”Participants” (page 4) lines 4-5: ”Before questionnaire distribution, subjects’ intention was inquired. What is meant by the ”subjects’ intention” by the authors here? Is this a reference to a voluntary agreement to participate in research? If so, then informed consent would be the proper term.

 Ans:  Thank you for the comment. We added “Before questionnaire distribution, subjects’ consent was inquired.”

(18)The paragraph ”Participants” (page 4): There are some awkward expressions used in this section when reporting the ”basic information” such as: ”are the most”; ”is the most” that should be corrected.

  Ans:  Thank you for the comment. We rewrote the section and corrected some grammar errors.

(19)In what way (if any) are the basic information/demographic information given in this section and in Table 1 (page 5) made use of in this study. This should at least be reflected upon in the discussion section of the manuscript (for example, how this data could have been used or could be used in similar future studies or can possibly be used by the authors in the future).

(IV) Measurement

(20) Table 1 (page 5): Please correct ”One children” to ”One child”

  Ans:  Thank you for the comment. Correction was made accordingly.

(21)The paragraph “Sports Participation (page 6): The first sentence “Sports participation aims to test participants’ average weekly frequency to participate in sports (frequency), average participation time of each time (participation time) and sports intensity”. The phrases in parentheses (frequency) and (participation time) seem redundant in this sentence. Maybe the sentence could read: Sports participation aims to test participants’ average weekly frequency to participate in sports, average participation time of each sports exercise, and sports intensity-

 Ans:  Thank you for the comment. We rewrote this part and cited Professor Fox’s originally publication related to this issue.

(22)In this same paragraph, the authors refer to Fox’s study: “According to formula applied by Fox [32] sports participation degree = frequency X (sports intensity + participation time).” This study referenced by the authors has no explicit formula of such kind presented. Fox does speak about “different formulas of frequency, intensity and duration of exercise apply for different mechanisms and perhaps different populations.” (Fox 1999, 416), but seems not to offer any explicit formula to which the authors in this manuscript refer to. Authors, please provide an explanation/revision to this or a rebuttal.

  Ans:  Thank you for the comment. We rewrote this part and cited Professor Fox’s originally publication related to this issue.

(V) Results

(23)When reporting the results (page 6) “The structural model and hypothesis testing” there could be added – especially in the case of H4 and H5 – “statistically insignificant”. Mere categories of significant and insignificant are weaker.

Ans:  Thank you for the comment.  We rewrote this by inserting a table and explaining which variable is the most predicting variable for sport participation.

(24)Paragraph “Explanatory Power” (page 7): Here the authors state that the predictive power of their research model can explain 38% of the variance of women’s sports participation. In my opinion, the authors should more clearly in this paragraph state that the agents of socialization of community and mass media are in themselves statistically insignificant and how that relates to this models’ explanatory power. Please, check also the grammar in this paragraph.

Ans:  Thank you for the comment. We deleted this section and did the explanation in the previous section.

(VI) Discussion

(25)First paragraph (page 8): ”Therefore, women with higher internal locus of control are not afraid of frustration and pressure in sports learning process.” The word ”therefore” here used by the authors refer to some conclusion. It is somewhat unclear, however, where this conclusion is drawn from: from this study’s results, from other studies mentioned just before this therefore-sentence. Furthermore, and perhaps even more importantly the authors should ponder the legitimacy of their conclusion or at least state this differently.

Ans:  Thank you for the comment. We rewrote the section and avoid “therefore”. One reference seemed not to fit in this section so we deleted it.

(26)Second paragraph (page 8): Why are the agents of socialization presented in the order of ”family and friends” while the results indicate the order of friends and family?

Ans:  Thank you for the comment. We changed the order accordingly.

(27)Second paragraph (page 8): ”It supports that individuals’ sports participation is accomplished by agents of socialization [39].” This sentence should be rephrased: ”it supports” should take note that these are the results of this study; the reference [39] is a study conducted with young children – this should also become clear from the text.

Ans:  Thank you for the comment. We changed the expressions to “It suggested that women’s sports participation level can be enhanced” and reference[39] was deleted because of not highly related to our study subjects.

(28)Second paragraph (page 8): The authors state that  “the finding of this study suggests that support of family and friends positively influences women’s sports participation.” In the very next sentence, they refer to the study conducted by Coleman, Cox, and Roker [40] and state that the “friendship group is the most critical factor of sports participation”. Maybe the order of presentation in the first sentence should be reversed to “friends and family” instead of “family and friends” because that is the order of the predictive power of friends (.29) and family (.10). This would be more coherent.

 Ans:  Thank you for the comment. We changed the order accordingly.

(29)The last whole paragraph on page 8: There are some English editing needed here: i.e., “it encouraged those who could walk to the place to practice exercises”; “For instance, when lighting of sports places is sufficient and width of path is appropriate (please specify what is meant with the “width of path” in this particular instance) ; “Therefore, in order to strengthen women’s sports participation, it must attempt to enhance (to whom does “it” refer to).

Ans: Thank you for the comment. We replaced “width of path” with “width of running tracks” and we repharsed the sentences “Therefore, in order to strengthen women’s sports participation, it must attempt to enhance” into “Therefore, we suggest the enhancements of environmental atmosphere, convenience and safety to promote women’s sports participation”.

(30)A more general comment: The whole discussion section could be re-arranged so that the structure follows according to the individual correlations and impacts on women’s sports participation (i.e., (1) Sport Socialization Situations… I leave this to the authors’ discretion.

 Anyhow, the significance and “impact factors” should be clear in the text. Furthermore, there should be some deeper contemplation upon why within this research setting and based upon this data the significances of community and mass media stayed statistically insignificant (although this topic is mentioned in Recommendations)

(31)Another more general comment: In this current manuscript there is very little discussion regarding the limitations of this study as well as the possible problems of (external) validity and generalizability of the results presented by the authors. Some notions regarding the most common possible pitfalls of questionnaire-based studies could be added.

Especially, the generalizability aspect of this study’s results should be reflected upon and explicated here and also in the recommendations section (i.e., How well are the results representative of larger population/other countries/contexts, etc.? What are the limitations of the results and recommendations of this study?).

  Ans: Thank you for the comment. The reliability of each construct was reported in the measurement section. As for the population, we reported in the participants.

(VII) Recommendation

(32) It would perhaps be more logical to present the recommendations in the order of relevance based on the findings of this study, at least inside the categories; currently, the authors write (page 8) “agents of socialization of family, friends and sport socialization are key factors…”.  So, sport socialization situations are the most important key factor, second most important, from the category of agents of socialization are friends and then family, and locus of control come before parents. Therefore, the authors should consider revising the order of presentation here.

Ans: thank you for comments. We had rearranged the order.

(33)Furthermore, “the locus of control” aspect is not very clearly presented in the recommendations section. This should be revised because of its importance for sports participation according to the findings presented by the authors.

Ans: thank you for comments. We added “Locus of control is another factor affecting women’s sports participation. Therefore, we recommend the government to promote equal education opportunities and encourage women to engage in sports to increase women’s internal locus of control toward sports participation.”

(34)The last paragraph (page 9): “It is different from some related research”. Here the authors speak about the role of community and mass media with respect to women’s sports participation. Some references to what these “some related research” are, is required by the authors.

Some recommended resources are: Tsai’s (2009) Media systems and their effects on women's sport participation in Taiwan. Sport, Education and Society, 14:1, 37-53, DOI: 10.1080/13573320802615023 ; Chen et al. (2004) Cultural and Social Factors Affecting Women’s Physical Activity Participation in Taiwan. Sport, Education and Society, 9(3), 379-393  

Ans: Thank you for the suggestions. We added some references here

Reviewer 3 Report

Abstract: English grammar needs to be fixed

Intro: Missing prevalence/incidence rates for Taiwan and would expect them there (civil illness) and also for sports participation. The first sentence is quite pessimistic. the introduction also contains several incomplete sentences. 'prevention of disease and control': this statement does not make sense. 

2nd paragraph: 'including important others' think you mean significant social support (role models). 

Your last paragraph may have a cultural view - there are big differences and high participation rates in western countries. What might be the cultural aspects that might not be supporting in Taiwan?  

The introduction is lacking overview of the theoretical concepts used for the study. I.e. LOC/Self-Efficacy and social support (rolemodelling, avaiability and meaning of opportunities). what happens if there are not any clubs/schools org  in relation to mental health? 

The hypothesis (H1) needs some clarification. Higher LOC will predict participation? 

H2-h5: can be combined to: Factors of Socialization can promote sport participation in females. 

Methods: 'subject intentions was inquired': informed consent was acquired. the descriptive statistics are in the chart and need not be repeated in text. Missing age means and standard deviation of the sample. 

Results: Missing all descriptives with means and standard deviations for all scales. There are several t-tests that are to be expected (Work, sports participation, prior school sport participation)

that may be relevant. 

Discussion: Needs to be rearranged. The hypothesis H1 was expected to be more predictive (how the paper was written up) but other variables were shown to be more important. The discussion is just a sequential explanation of the results and not a discussion of the impact of the variables taking into consideration previous findings. 

This paper is an interesting contribution especially for cross cultural factors but needs some clarifications. How do females in Taiwan compare to other female populations? Are there common factors? These aspects need to be introduced and discussed. 

Author Response

Abstract: English grammar needs to be fixed

Intro: Missing prevalence/incidence rates for Taiwan and would expect them there (civil illness) and also for sports participation. The first sentence is quite pessimistic. the introduction also contains several incomplete sentences. 'prevention of disease and control': this statement does not make sense. 

 Ans: Thank you for the comment. We rewrote this section and added some other references.

2nd paragraph: 'including important others' think you mean significant social support (role models). 

 Ans: Thank you for the comment. We replaced “important others” with “significant others”

Your last paragraph may have a cultural view - there are big differences and high participation rates in western countries. What might be the cultural aspects that might not be supporting in Taiwan?  

 Ans: Thank you for the comment. We deleted the sentences.

The introduction is lacking overview of the theoretical concepts used for the study. I.e. LOC/Self-Efficacy and social support (rolemodelling, avaiability and meaning of opportunities). what happens if there are not any clubs/schools org  in relation to mental health? 

The hypothesis (H1) needs some clarification. Higher LOC will predict participation? 

 Ans: Thank you for the comment. We rewrote the section and added some references to support our hypotheses.

H2-h5: can be combined to: Factors of Socialization can promote sport participation in females. 

 Ans: Thank you for the comment. We rewrote those sections.

Methods: 'subject intentions was inquired': informed consent was acquired. the descriptive statistics are in the chart and need not be repeated in text. Missing age means and standard deviation of the sample. 

 Ans: Thank you for the comment. We rewrote those sections.

Results: Missing all descriptives with means and standard deviations for all scales. There are several t-tests that are to be expected (Work, sports participation, prior school sport participation) that may be relevant. 

 Ans: Thank you for the comment. Summary statistics of construct variables was added.

Discussion: Needs to be rearranged. The hypothesis H1 was expected to be more predictive (how the paper was written up) but other variables were shown to be more important. The discussion is just a sequential explanation of the results and not a discussion of the impact of the variables taking into consideration previous findings. 

 Ans: Thank you for the comment. We rewrote the paragraphs.

This paper is an interesting contribution especially for cross cultural factors but needs some clarifications. How do females in Taiwan compare to other female populations? Are there common factors? These aspects need to be introduced and discussed.

Round 2

Reviewer 1 Report

APA format for titles

Titles and subtitles are back to back

Conclusions speak to environmental factors though these factors were intentionally excluded from the study.

The information in the background is relevant but presented in a disjointed manner. 

Author Response

Titles and subtitles are back to back

Ans: We revised that as well.

Conclusions speak to environmental factors though these factors were intentionally excluded from the study.

Ans: We added it back to discuss all factors.

The information in the background is relevant but presented in a disjointed manner. 

Ans:  we revised these as well. Thank you

Reviewer 2 Report

Thanks to the authors for the revised manuscript. There are some important considerations below:

General comments:

Check references: Multiple reference numbers are way off. Check and correct these!

English language: While there are some improvements in this revised manuscript, the English language is still in obvious need of language editing/proofreading. There are simply too much grammar errors (i.e., subject/verb agreement, article use (definite/indefinite), verb tenses), some incomprehensible sentences, and linguistic structures present.

Questionnaire: It would be strongly advisable for the authors to include their questionnaire along with the manuscript (as an Appendix or in the form of Supplement material). This is, in my opinion, in accordance with the universal scientific principles and good conduct: repeatability and transparency, etc.

This went unanswered by the authors. Please provide a reason/rebuttal why the questionnaire used cannot be as an Appendix file etc. or is not necessary as such.

More specific comments (few are echoed from Round 1 as they are not adequately handled)

(1)    Introduction (page 1): ”It reveals that regularly participations in sports and psychological therapy can improve psychological symptoms”. (Bolding for grammar issues). Why is psychological therapy introduced here? Furthermore, the next sentence jumps back to purely physical aspects of health. Authors, please make this more understandable and coherent.

(2)    Introduction (page 2): I cannot seem to locate the publication or the journal behind reference [8]. The authors answered that they had ”deleted the sentences and the related reference” but there it is still now as [8] (original manuscript: reference [5])

(3)   The third paragraph of the introduction (page 2) mentions the negative effects of gender stereotypes and patriarchal thinking on women’s sports participation. Do these reflections surface in the discussion section of the manuscript? Furthermore, are there any findings from this study that could shed some light on these issues mentioned.

 Ans: Thank you for the comment. We rewrote the sentences.

I would at least advice the authors to change the wording in the sentence ”In sports field there was also the inequity caused by traditional gender concept”;  stereotype and/or role(s) would be better than gender concept.

(4)   The authors write: “Subjects of this study are all women. Therefore, agents of socialization refer to family, friends, mass media and community.” Why and on what grounds do the authors choose or narrow as the agents of socialization in women particularly these four (family, friends, mass media and community? How is this decision justified (i.e., do men have some other agents of socialization)? At least, the “therefore”, starting the second sentence seems to indicate that the four agents of socialization are specifically and exclusively pertaining only women.

 Ans: Thank you for the comment. Agents of socialization include family, institutional agent (school, workplace, and government), friends, mass media and community. Since the interest of our study topic is related to sports participation of female adults, we had excluded the effect of institutional agent. Therefore, agents of socialization here in the study include family, friends, mass media and community.

The effort of the authors to explicate the agents of socialization is good. They also clarify that within this study (on female adults) they have ”excluded the effect of institutional agent [school, workplace, and government]”. Still, this does not explain why the institutional aspect of socialization is irrelevant to female adults. The authors themselves state on page 10 that ”As for sport socialization situations, we suggest related governmental or administrative sectors can plan sports places for women…” and later on in relation to LOC that ”we recommend the government to promote equal education opportunities and encourage women to engage in sports…” So this might be something to think about.

Furthermore, what is the motivation behind the governmental promotion of ”equal education opportunities” when the authors state (page 7) that ”Among the background variables of women about sports participation, no significant differences were found between marriage, LEVEL OF EDUCATION, … There were significant differences … participate school sport team during student hood”. So, there is no moderating significance between sports participation and level of education (which seems to undermine the very idea of government’s promotion of equal education opportunities) but there is significance between sports participation and school sport team participation (which seems to undermine the exclusion of the ”institutional agent of socialization” in adult females.

(5)    Page 3: What does this sentence mean(?): ”Continuation of sport field requires of personal efforts and may also be affected by the interactions with people and environment involved” (Grammar)

(6)    Page 3: The sentence ”It influenced the promotion of community residents’ sports fashion.” seems somewhat misplaced and out of context.

(7)    Hypotheses H2-H5: Get rid of the ”Agent of” -beginnings. Let them simply read as:  H2: Family has…; H3: Friends has…; H4: Community has…; H5: Mass media has… Also, don’t use the suffix -ly with significant (that is significantly), it doesn’t make sense in English.

(8)    Method – Participants (page 4): There are some issues remaining here: (a) ”were married 85.9%” (should this rather be 86.1 %); (b) ”(US$2600)” (why the use parenthesis?); (c) ”as to full time job … positive responses (79%) (should this rather be 93%); (d) why the age is mentioned to times ”between 20-50 years old” and ”as to age, 31-40 is the most”; (e) and why there is still two ”is the most expressions here, while they should be gone?

(9)    Page 5: ”People of externals…” (Grammar)

(10) Page 6: Level of Exercise Participation: There are three instances of the word ”responder” used. I believe the authors mean ”respondent”.

(11) Page 6: Level of Exercise Participation: What does the following question mean: ”how often do you feel after workout?”

(12) Pages 6-7 ”Descriptive statistics”: Some grammar issues ”is above general to agree”; ”As for sports participant”; ”student hood”

(13) Page 8 ”Discussion”: ”it seems obviously” (grammar)

(14) Page 8 ”Discussion”: ”In related to sports field, research found teams’ players had higher positive perception on coaches’ leadership” (The original manuscript included the notion that ”when coaches have internal control, the players show higher positive perception on coaches’ leadership.)

(15) Page 9: First sentence of the second paragraph ”Effect of agents of socialization of friends and family… Few lines down: ”The finding of this study suggests that support of family and friends…” Be coherent with the order of presentation.

(16) Page 9: ”The result supports finding of Coleman, Cox and Roker…” Is the intention of the authors here to rather say that ”These results (namely, ”the support of friends and family positively influences women’s sports participation) are supported by previous studies…” Then the authors could go along and list the studies of Coleman et al.; Young et al.; Sharma et al.

(17) Page 10: Regarding the non-significance of the community and mass media to women’s sports participation in Taiwan the added references are good. There could still be some considerations/educated guesses as to why these findings differ ”from some related research”.

Author Response

General comments:

Q:Check references: Multiple reference numbers are way off. Check and correct these!

A: Thank you. We made correction accordingly.

English language: While there are some improvements in this revised manuscript, the English language is still in obvious need of language editing/proofreading. There are simply too much grammar errors (i.e., subject/verb agreement, article use (definite/indefinite), verb tenses), some incomprehensible sentences, and linguistic structures present.

Q: Questionnaire: It would be strongly advisable for the authors to include their questionnaire along with the manuscript (as an Appendix or in the form of Supplement material). This is, in my opinion, in accordance with the universal scientific principles and good conduct: repeatability and transparency, etc.
This went unanswered by the authors. Please provide a reason/rebuttal why the questionnaire used cannot be as an Appendix file etc. or is not necessary as such.

Ans: Thank you. We had provided the questionnaire in appendix for reference

More specific comments (few are echoed from Round 1 as they are not adequately handled)

(1)    Introduction (page 1): ”It reveals that regularly participations in sports and psychological therapy can improve psychological symptoms”. (Bolding for grammar issues). Why is psychological therapy introduced here? Furthermore, the next sentence jumps back to purely physical aspects of health. Authors, please make this more understandable and coherent.

It reveals that regularly participations in sports and psychological therapy can improve psychological symptoms.
Ans: Thank you for the comment. Originally, we wanted to point out the benefits of sports participation but also call on people’s attention for psychological therapy if necessary due to the decease of our former department head who believed regular participation in sports solely could be done with medication or therapy (He died of myocardial infarction and was suffered from high blood pressure without medication). Anyway, since it may distract readers’ focus on sports participation, we decided to delete this claim.

(2)    Introduction (page 2): I cannot seem to locate the publication or the journal behind reference [8]. The authors answered that they had ”deleted the sentences and the related reference” but there it is still now as [8] (original manuscript: reference [5])
Ans: Thank you. We had provided the corresponding reference.
(3)   The third paragraph of the introduction (page 2) mentions the negative effects of gender stereotypes and patriarchal thinking on women’s sports participation. Do these reflections surface in the discussion section of the manuscript? Furthermore, are there any findings from this study that could shed some light on these issues mentioned.

Ans: Actually, we didn’t discuss much of this since the article focuses on the factors which may affect women’s sports participation. As for cross cultural factors, we haven’t touched that far and will consider to to it for the next step. However, it is obvious Taiwanese women’s sports participation was strongly influenced by western women’s sport atmosphere and increasing.

I would at least advice the authors to change the wording in the sentence ”In sports field there was also the inequity caused by traditional gender concept”;  stereotype and/or role(s) would be better than gender concept.
Ans: thank you for the comment. We made the change accordingly.

(4)   The authors write: “Subjects of this study are all women. Therefore, agents of socialization refer to family, friends, mass media and community.” Why and on what grounds do the authors choose or narrow as the agents of socialization in women particularly these four (family, friends, mass media and community? How is this decision justified (i.e., do men have some other agents of socialization)? At least, the “therefore”, starting the second sentence seems to indicate that the four agents of socialization are specifically and exclusively pertaining only women.
Ans:  thank you for the comment. We added “
Specifically, after adolescence, institutional agent plays much less influence on each individual’s sports participation”

“Ans: Thank you for the comment. Agents of socialization include family, institutional agent (school, workplace, and government), friends, mass media and community. Since the interest of our study topic is related to sports participation of female adults, we had excluded the effect of institutional agent. Therefore, agents of socialization here in the study include family, friends, mass media and community.”

The effort of the authors to explicate the agents of socialization is good. They also clarify that within this study (on female adults) they have ”excluded the effect of institutional agent [school, workplace, and government]”. Still, this does not explain why the institutional aspect of socialization is irrelevant to female adults. The authors themselves state on page 10 that ”As for sport socialization situations, we suggest related governmental or administrative sectors can plan sports places for women…” and later on in relation to LOC that ”we recommend the government to promote equal education opportunities and encourage women to engage in sports…” So this might be something to think about.

Ans: thank you for the comment. Well, there may be some misunderstandings here. The statement here was to call out the attention of providing women exclusive sport filed from the government. As you can see that the influence of the government on women’s sports participation is global not a personal effect, that was why we didn’t included the governmental effect on each individual.

Furthermore, what is the motivation behind the governmental promotion of ”equal education opportunities” when the authors state (page 7) that ”Among the background variables of women about sports participation, no significant differences were found between marriage, LEVEL OF EDUCATION, … There were significant differences … participate school sport team during student hood”. So, there is no moderating significance between sports participation and level of education (which seems to undermine the very idea of government’s promotion of equal education opportunities) but there is significance between sports participation and school sport team participation (which seems to undermine the exclusion of the ”institutional agent of socialization” in adult females.

Ans: thank you for the comment. In oriental society, mostly, the emphasis on academic success is the most important thing. The importance of physical education was less stressed. Therefore, there were not many sports club on elementary schools or high schools not even mentioned the participation of sports club. Fortunately, the emphasis on physical activities had increased. Therefore, there was no big surprise in that there was no significance of sports participation among levels of education.

(5) page 3: What does this sentence mean(?): ”Continuation of sport field requires of personal efforts and may also be affected by the interactions with people and environment involved” (Grammar)

Ans: thank you for the comment. We deleted this sentence.

(6)    Page 3: The sentence ”It influenced the promotion of community residents’ sports fashion.” seems somewhat misplaced and out of context.
Ans: thank you for comment. We rephrased the sentences into “
The accessibility, convenience of sport field in community may also increase people’s sports participation intention.”

(7)    Hypotheses H2-H5: Get rid of the ”Agent of” -beginnings. Let them simply read as:  H2: Family has…; H3: Friends has…; H4: Community has…; H5: Mass media has… Also, don’t use the suffix -ly with significant (that is significantly), it doesn’t make sense in English.

Ans: Thank you for comment. We made changes accordingly.

(8)    Method – Participants (page 4): There are some issues remaining here: (a) ”were married 85.9%” (should this rather be 86.1 %); (b) ”(US$2600)” (why the use parenthesis?); (c) ”as to full time job … positive responses (79%) (should this rather be 93%); (d) why the age is mentioned to times ”between 20-50 years old” and ”as to age, 31-40 is the most”; (e) and why there is still two ”is the most expressions here, while they should be gone?

Ans: Thank you for the comments. (a)Among all participants, 71.9% were married. 85.9% were between 20~50 years old.(b) parentheses were deleted (C) we corrected theses.

(9)    Page 5: ”People of externals…” (Grammar)
Ans: Thank you for comment.We changed it to “
people with higher external locus of control”

(10) Page 6: Level of Exercise Participation: There are three instances of the word ”responder” used. I believe the authors mean ”respondent”.

Ans: Thank you for the comment. We corrected these errors.

(11) Page 6: Level of Exercise Participation: What does the following question mean: ”how often do you feel after workout?”

Ans: Thank you for the comment. The correct expression should be “
In average, how did you feel after workout?”

(12) Pages 6-7 ”Descriptive statistics”: Some grammar issues ”is above general to agree”; ”As for sports participant”; ”student hood”
Ans: thank you for comments. (a)We changed ”is above general to agree” to “was high”
(b)
we changed the sentences to “whether to join sport clubs at present or previous experience in participating school sport team during school life.”

(13) Page 8 ”Discussion”: ”it seems obviously” (grammar)
Ans: Thank you for the comment. We changed it to “
it seemed that”

(14) Page 8 ”Discussion”: ”In related to sports field, research found teams’ players had higher positive perception on coaches’ leadership” (The original manuscript included the notion that ”when coaches have internal control, the players show higher positive perception on coaches’ leadership.)
Ans: Thank you for the comment. We added it back ”
when coaches have internal control”

(15) Page 9: First sentence of the second paragraph ”Effect of agents of socialization of friends and family… Few lines down: ”The finding of this study suggests that support of family and friends…” Be coherent with the order of presentation.
Ans: Thank you for the comment.We changed that.

(16) Page 9: ”The result supports finding of Coleman, Cox and Roker…” Is the intention of the authors here to rather say that ”These results (namely, ”the support of friends and family positively influences women’s sports participation) are supported by previous studies…” Then the authors could go along and list the studies of Coleman et al.; Young et al.; Sharma et al.
Ans: Thank you for the comment. We changed that.

(17) Page 10: Regarding the non-significance of the community and mass media to women’s sports participation in Taiwan the added references are good. There could still be some considerations/educated guesses as to why these findings differ ”from some related research”.
Ans: Thank you for the comments. We provided explanation in the discussion section. (
On the other hand, effects of community and mass media on women’s sports participation are not significant. It seemed the support from community (nearby sport field, sport-related event and neighbors’ company) for sports participation was not strong enough. Also, support from the mass media was not significant indicating the necessity of posting more women-oriented sports-related agenda, events and advertisements.

Reviewer 2:
This paper is an interesting contribution especially for cross cultural factors but needs some clarifications. How do females in Taiwan compare to other female populations? Are there common factors? These aspects need to be introduced and discussed.

Ans: thank you for comments. The article focuses on the factors which may affect women’s sports participation. As for cross cultural factors, we haven’t touched that far and will consider to to it for the next step. However, it is obvious Taiwanese women’s sports participation was strongly influenced by western women’s sport atm

Reviewer 3 Report

The authors have done substantial changes to improve the manuscript. 

Author Response

The authors have done substantial changes to improve the manuscript

Ans: Thank you